# Combinations of Lemongrass and Star Anise Essential Oils and Their Main Constituent: Synergistic Housefly Repellency and Safety against Non-Target Organisms

**DOI:** 10.3390/insects15030210

**Published:** 2024-03-20

**Authors:** Mayura Soonwera, Jirisuda Sinthusiri, Hataichanok Passara, Tanapoom Moungthipmalai, Cheepchanok Puwanard, Sirawut Sittichok, Kouhei Murata

**Affiliations:** 1Department of Plant Production Technology, School of Agricultural Technology, King Mongkut’s Institute of Technology Ladkrabang, Ladkrabang, Bangkok 10520, Thailand; hataichanok.pa@kmitl.ac.th (H.P.); 64604012@kmitl.ac.th (T.M.); 63604010@kmitl.ac.th (C.P.); best_pest22@hotmail.com (S.S.); 2Community Public Health Program, Faculty of Public and Environmental Health, Huachiew Chalermprakiet University, Bang Phli, Bang Chalong 10540, Thailand; jiri_ja@yahoo.com; 3Office of Administrative Interdisciplinary Program on Agricultural Technology, School of Agricultural Technology, King Mongkut’s Institute of Technology Ladkrabang, Ladkrabang, Bangkok 10520, Thailand; 4School of Agriculture, Tokai University, Kumamoto 862-8652, Japan; kmurata@agri.u-tokai.ac.jp

**Keywords:** housefly, essential oil combination, synergistic repellent, lemongrass essential oil, star anise essential oil, geranial, *trans*-anethole, non-target species

## Abstract

**Simple Summary:**

The housefly (*Musca domestica*) is a global vector of several pathogens and a nuisance to humans and animals. Repellents reduce the risk of housefly vector diseases. Green repellents from plant essential oils (EOs) and their active compounds were, at the time, a promising option against houseflies. This study evaluated the housefly-repellent activities of every tested EO, their main constituents, and several combinations of them. The combinations were lemongrass (*Cymbopogon citratus*) EO + *trans*-anethole, lemongrass EO + star anise (*Illicium verum*) EO, geranial + *trans*-anethole, and star anise EO + geranial. The efficacies of every formulation were compared against each other and DEET (a synthetic repellent). All combinations were more effective in repellency than all single-component formulations and DEET. More significantly, all of them were safe for four non-target species: guppy (*Poecilia reticulata*), molly (*Poecilia latipinna*), dwarf honeybee (*Apis florea*), and stingless bee (*Tetragonula pagdeni*). These combinations could be developed into valuable green repellents for housefly vector disease management.

**Abstract:**

The present study evaluated the housefly repellency of single-component formulations and combinations of lemongrass and star anise essential oils (EOs) and their main constituents. The efficacies of the combinations were compared against those of single-component formulations and DEET. Safety bioassays of all formulations and DEET on non-target species—guppy, molly, dwarf honeybee, and stingless bee—were conducted. GC–MS analysis showed that the main constituent of lemongrass EO was geranial (46.83%) and that of star anise EO was *trans*-anethole (92.88%). All combinations were highly synergistic compared to single-component formulations, with an increased repellent value (IR) of 34.6 to 51.2%. The greatest synergistic effect was achieved by 1.0% lemongrass EO + 1.0% *trans*-anethole combination, with an IR of 51.2%. The strongest, 100% repellent rate at 6 h was achieved by 1.0% geranial + 1.0% *trans*-anethole. They were twice as effective as DEET and caused obvious damage to housefly antennae under microscopic observation. All single-component formulations and combinations were benign to the four tested non-target species. In contrast, DEET was highly toxic to them. The synergistic repellency and biosafety of these two combinations are compellingly strong support for developing them into an effective green repellent.

## 1. Introduction

The housefly, *Musca domestica* L. (Muscidae), is a global nuisance and vector pest of humans, domestic animals, poultry, and livestock [1,2]. This pest has become a worldwide problem and acts as a mechanical vector of several severe pathogens that cause serious diseases, such as food-born illnesses, leprosy, and typhoid in humans and avian flu, fowl cholera, anthrax, and porcine reproductive and respiratory syndrome diseases in livestock [2,3]. The economic loss per year due to housefly problems was estimated at more than USD 1 billion in 2021 in the United States [1,4,5]. The effective management of housefly populations is very complicated and difficult, especially in areas with abundant houseflies that are difficult to access [5,6]. Hence, synthetic insecticide control became the first and most popular option for housefly management [5,6]. The most popular synthetic repellent for insect vector control was *N*,*N*-diethyl-*m*-toluamide (DEET) [6,7]. Unfortunately, DEET, acting directly as an insect repellent, was found to be highly toxic to the human nervous system and non-target species [8,9].

At this time, a renewed interest in green insecticides and repellents as alternative insect management tools was spurred by the global problem of housefly resistance to synthetic insecticides and synthetic repellents, their negative effects on non-target organisms, and the ecological pollution and imbalance that they have caused [8,9]. Green repellents from plant essential oils (EOs) and their major constituents have the potential to be safe alternative agents for preventing houseflies that have not been killed by synthetic agents in sensitive areas such as hospitals, nurseries, and nursing homes [6,10]. To summarize, plant EO-based repellents are highly effective, highly species-specific, biodegradable, and safe for humankind, making them a promising option for good housefly management [7,10,11].

Currently, there are many reports presenting EOs showing repellency against housefly adults: lemongrass (*Cymbopogon citratus*), peppermint (*Mentha piperita*), bergamot mint (*Mentha citrata*), blue gum (*Eucalyptus globulus*), khus grass (*Vetiver zizanoides*), and turmeric (*Curcuma longa*) by filter paper application [12]; tea tree (*Melaleuca alternifolia*), Indian tree basil (*Ocimum gratissimum*), breckland thyme (*Thymus serpyllum*), star anise (*Illicium verum*), nutmeg (*Myristica fragrans*), mango ginger (*Curcuma amada*), and sweet flag (*Acorus calamus*) by spray application [7,13]; and cinnamon (*Cinnamomum zeylanicum*) by cotton pad application [14].

Moreover, an EO combined with its major ingredient or with another EO showed synergistic repellency against housefly adults. Those combinations include peppermint EO + lemongrass EO, peppermint EO + sweet orange (*Citrus sinensis*) EO, and peppermint EO + blue gum EO [6,15]. Combining EOs with their major constituents increased their insecticidal and repellent activities over those of single EOs. Combinations of nutmeg EO + geranial, α-pinene + geranial, and geranial + *trans*-anethole exhibited strong insecticidal synergy against housefly adults in spray form [16]; a combination of star anise EO + cinnamon (*Cinamomum verum*) EO exhibited highly repellent and ovicidal effects in spray form against the American cockroach, *Periplaneta americana* [17]. A combination of ylang-ylang (*Cananga odorata*) EO + lemongrass EO showed repellent activity in spray form against mosquito females of *Aedes aegypti* and *Culex quinquefasciatus* [18]. Furthermore, several combinations of EOs and their main constituents showed high toxicity against target species (*Ae*. *aegypti* and *Ae*. *albopictus* eggs) but were non-toxic to the tested non-target predators (*Poecilia latipinna* and *Poecilia reticulata*) [19].

Several combinations of EOs and their major constituents had high potential for use as green repellents against houseflies in that they were, at the time, more effective than DEET, a popular synthetic repellent, and they were safer repellents for humans, environmentally friendly, and non-toxic to non-target predators [19,20]. Consequently, this study was motivated to investigate the adult housefly and the repellency incurred by single-component and combinations of lemongrass and star anise EOs and their main active constituents. These two EOs were selected because they have been widely reported to possess medicinal, antibacterial, and insecticidal properties, as well as to be safe for humans and the environment [16,19]. The synergistic housefly repellency of the combinations and their biosafety against non-target species were evaluated. The non-target species were two pollinators, *Apis florea* F. (dwarf honeybee) and *Tetragonula pegdeni* Schwarz (stingless bee) (both in the Apidae family) and two aquatic predators, *Poecilia latipinna* Lesueur (molly) and *Poecilia reticulata* Peters (guppy) (both in the Poeciliidae family). These four non-target species are common pollinators and aquatic predators in Asia, including Thailand [19,20]. Once their synergistic repellent activity and safety for the four non-target species have been demonstrated, some of these EO formulations may be deemed effective and sustainable green repellent alternatives to DEET.

## 2. Materials and Methods

### 2.1. Plant Materials and Extraction of Two Plant Essential Oils

The parts of plants from which EOs in this study were extracted were fresh stems of lemongrass (*Cymbopogon citratus*, KMITL-01-11) and dried fruits of star anise (*Illicium verum*, KMITL-01-18). The collection of plant materials was conducted in compliance with national, regional, and international laws. Under the permit KD2021/002 of King Mongkut’s Institute of Technology Ladkrabang (KMITL), the authors were granted consent, sample monitoring rights, and authorization for the collection of plant materials. Fresh stems of lemongrass were collected from an organic farm in Bang Nam Priao, Chachoengsao Province, Thailand (latitude 13°53′44.44″ N and longitude 101°1′33.344″ E) from March–April 2023. Dried fruits of star anise were purchased from a Chinese herb supplier in Chachoengsao Province, Thailand. The two plant species were identified by Hataichanok Passara, a botanist from the KMITL herbarium at the School of Agricultural Technology of the same institute. This herbarium housed two plant specimens (KMITL-01-11 and KMITL-01-18) for future use.

EO extraction was performed by hydro-distilling 800 g of fresh stems of lemongrass and 800 g of dried fruits of star anise in 2000 mL of distilled water. After 3 h, each EO was filtered into a 100 mL brown bottle and stored at 4 °C until they were later used in repellent assays and gas chromatography–mass spectrometry (GC–MS) King Mongkut’s Institute of Technology Ladkrabang (KMITL), Bangkok, Thailand analysis.

The GC–MS system analyzed the composition in three replicates. The systems were an electron ionization system with 70 eV electron energy (30–500 m z^−1^) and an Agilent 6890-N gas chromatograph coupled to a 5973-N mass spectrometer with an HP-5 MS fused silica capillary column (30 m × 0.25 mm ID with 0.25 m film thickness of 5% phenyl-methylpolysiloxane coating). Each EO sample was diluted at a split ratio of 1:100 in ethyl alcohol, and 0.2 µL of the diluted sample was injected into the column. The oven temperature was set at 50 °C for 2 min, then raised at a rate of 10 °C min^−1^ to 200 °C and kept there for 3 min. The final step was to raise the temperature at a rate of 15 °C min^−1^ to 260 °C and keep it there for 20 min. The injector and detector temperatures were set at 260 °C. The retention times of all chemical constituents of each EO were analyzed and compared to those of the standard *n*-alkanes. Their retention index was compared with the reference values from Adams [21] or the National Institute of Standards and Technology (NIST) [22] or other sources in the published literature [23,24].

### 2.2. Chemicals

Technical grade 96% geranial (CAS 5392-40-5, the major compound of lemongrass EO) and 98.5% *trans*-anethole (CAS 4180-23-8, the major compound of star anise EO) were purchased from Sigma-Aldrich Company Limited, Saint Louis, MO, USA. The two compounds were used to prepare 70% *v*/*v* stock solutions in ethyl alcohol (purchased from T.S. Interlab Company Limited, Bangkok, Thailand). The 12% *w*/*w* DEET (Softfell^®^) used as the positive control was purchased from CP Consumer Products Company, Minburi, Bangkok, Thailand.

### 2.3. The Tested Formulations

The suitable strength or concentration of EOs to be tested was based on our previous studies [16,25]. At these assigned concentrations, these EO formulations were already shown to be effective against houseflies in terms of their ovicidal and insecticidal effects [17,19]. Therefore, the strength was chosen in anticipation of effective repellency while not chosen to be so strong as to be toxic to the non-target species. The solvent for diluting the EO extract was 70% *v*/*v* ethyl alcohol. The eight tested strengths of single-component formulations were lemongrass EO at 2.5% and 5.0%, star anise EO at 2.5% and 5.0%, geranial at 0.5% and 1.0%, and *trans*-anethole at 0.5% and 1.0%. The tested strengths for combinations were 1.25% lemongrass EO + 1.25% star anise EO, 2.5% lemongrass EO + 2.5% star anise EO, 0.25% geranial + 0.25% *trans*-anethole, 0.5% geranial + 0.5% *trans*-anethole, 1.0% geranial + 1.0% *trans*-anethole, 1.0% lemongrass EO + 1.0% *trans*-anethole, and 1.0% star anise EO + 1.0% geranial.

### 2.4. Housefly Rearing

Larvae of housefly *M. domestica* were obtained from a housefly colony raised in the entomological laboratory of the School of Agricultural Technology, KMITL, under the conditions of 25.3 ± 1.5 °C, 70.5 ± 2.5% RH, and 13.0 h light and 11.0 h dark periods. They were reared with steamed mackerel mixed with milk powder at a ratio of 1:0.25 [25]. After 7–10 days, the larvae developed into pupae and adults. The adults were fed with 10% honey solution + milk powder + mineral water at a ratio of 5:5:90. Then, 3-to-4-day-old adults were collected to serve as test subjects for the repellent bioassay [25].

### 2.5. Repellent Activity Bioassay

Repellency against housefly adults was evaluated by putting them into two test cages (size 40 × 40 × 40 cm): the first cage contained a tested formulation, but the second cage did not contain a tested formulation. The test cage was designed especially for repellency bioassay. It was made of five plastic sheets (40 × 40 cm) and one screen sheet (35 × 35 cm) at the top of the cage, as can be seen in Figure 1. The two cages were connected by a rectangular hole (10 × 10 cm). When the repellency bioassay was conducted, the cage with a tested formulation was opened to the cage with no tested formulation through the interconnecting passage. Two milliliters of tested formulations were dropped onto a filter paper (Whatman No.1^®^, Cytiva Global Life Sciences Solutions Operations UK Ltd., Buckinghamshire, UK, 8.5 cm in diameter) which was then put on a petri dish in the cage containing the treatment, while two milliliters of ethyl alcohol were dropped onto a filter paper which was then put on a petri dish in the cage containing no tested formulation. Ten grams of steamed mackerel fish meat were placed on top of both filter papers, as food, to attract the houseflies. A wet filter paper was placed in the petri dish in each cage and had a little ball of cotton wool placed adjacent to it that was soaked with honey solution 10% + multivitamin syrup 2.0% to provide the subjects with food and water. Twenty-five three-day-old housefly adults were released into each cage. The positive control, 12% (*w*/*w*) DEET, was tested concurrently with the tested formulations. Each treatment was performed in 5 replicates at the same time. The whole test is summarized in Figure 1 below.

The repellent rates against housefly adults were observed and recorded at 1.0, 3.0, and 6.0 h. It was observed whether the houseflies landed on the filter papers on the petri dish in the cage with the tested formulation or in the cage with no tested formulation. The number of landings for at most 5 min after the end of each period were counted, starting from the start time to the end of 1.0, 3.0, or 6.0 h, following the protocol made by the Thai Industrial Standards Institute, Ministry of Industry [26]. The houseflies might land and then leave or stay on the filter paper until the end of each time period. The following formula was used to determine the adult repellent rate (RR) [26]:%RR = [A − B/A + B] × 100(1)
where A is the total number of housefly adults landing on the untreated filter paper, and B is the total number of housefly adults landing on the treated filter paper.

The repellent index (RI) was determined by the formula [19] below,
RI = %RR_tf_/%RR_DEET_,(2)
where %RR_tf_ is the %RR of the tested formulations, and %RR_DEET_ is the %RR of DEET.

RI < 1 means that the tested formulation was less potent than DEET, and RI > 1 means that the tested formulation was more potent than DEET.

The synergistic repellent index (SRI) was calculated by the following formula [19]
SRI = %RR_comb_/sum(%RR_sing_),(3)
where %RR_comb_ is the %RR of the combination, and %RR_sing_ is the (%RR of the single-component formulation.

SRI < 1 means that the combination was synergistic; SRI = 1 means that the combination was neither synergistic nor antagonistic; and SRI > 1 means that the combination was antagonistic.

The increased repellent value (%IR) was calculated by the following formula [16,25],
%IR = [%RR_comb_ − sum %RR_sing_/%RR_comb_] × 100,(4)
where %RR_comb_ is the %RR of the combination, and %RR_sing_ is the (%RR of the single-component formulation.

### 2.6. Safety Bioassay of Four Non-Target Species

#### 2.6.1. Bioassay of Dwarf Honeybee and Stingless Bee

The toxicity of the tested formulations was evaluated against two non-target pollinators (dwarf honeybee and stingless bee), following the methods of Pashte and Patil [27] and Chibee et al. [28]. One hundred adult dwarf honeybee and stingless bee workers were collected from the organic farm at the School of Agricultural Technology, KMITL.

The two non-target pollinators were identified by Jirisuda Sinthusiri, a qualified taxonomist, at Huachiew Chalermprakiet University, Thailand. One hundred workers of each species of pollinator were transferred into an insect cage (30 × 30 × 30 cm) and transported to the entomological laboratory within 1 h of the collection. All pollinator workers were fed with sugar solution 18.0% + multivitamin syrup 2.0% and maintained under the conditions of 25.5 ± 1.5 °C, 70.5 ± 2.5% RH, and 13.0 h light and 11.0 h dark periods for 2–3 days before the topical application test. The topical test consisted of applying 1 µL of each tested formulation to the mesonotum part (the dorsal of the second thorax) of each tested pollinator. After that, ten pollinators were transferred into a plastic box (10 × 10 × 5 cm) and fed with sugar solution 18.0% + multivitamin syrup 2.0%. Each treatment was performed five times, together with 12% (*w*/*w*) DEET positive control. The bee mortality was recorded at 1, 6, 12, 24, 48, and 72 h after exposure. The formulas below were used to determine the mortality rate (MR) of the bees:%MR = C/D × 100,(5)
where C is the number of dead bees, and D is the number of treated bees.

The safety index (SI) was determined using the formula below [16,19],
SI = LT_test_/LT_DEET_,(6)
where LT_test_ is the LT_50_ (50% Lethal Time value) of the tested formulation, and LT_DEET_ is the LT_50_ of DEET.

SI > 1 means that the tested formulation was safe for the bees, and SI < 1 means that the tested formulation was unsafe for the bees.

#### 2.6.2. Bioassay of Guppy and Molly

The toxicity of the tested formulations was tested against two non-target aquatic predators (guppy and molly), following the method of Moungthipmalai et al. [19]. Both fish predators were purchased from an organic farm in Minburi, Bangkok, Thailand. One hundred fish of each species were kept in a plastic container (40 × 60 × 30 cm) containing 50.0 L of clean water under the conditions of 25.6 ± 1.3 °C, 70.5 ± 2.5% RH, and 13.0 h light and 11.0 h dark periods. The concentration of each treatment was 10,000 ppm following [19]. Ten guppy or molly adults were put in a plastic container (35 cm in diameter and 18 cm in height) containing 5.0 L of clean water. Each tested formulation was applied at 10,000 ppm. Each treatment was tested five times, simultaneously with DEET. The mortality rate and abnormal behavior were recorded for 1, 3, 5, 7, and 10 days post-treatment. The safety index (SI) was determined by the formula [19] below,
SI = LT_tf_/LT_DEET_,(7)
where LT_tf_ is the LT_50_ of the tested formulation, and LT_DEET_ is the LT_50_ of DEET.

SI > 1 means that the tested formulation was safer than DEET for the non-target aquatic predator, and SI < 1 means that the tested formulation was less safe than DEET for the non-target aquatic predator.

### 2.7. Ethics and Guidelines for Bioassays

All bioassays in this study were approved by the Ethics committee of King Mongkut’s Institute of Technology Ladkrabang with the registration number KDS2021/002. They were performed per the ethical principles and guidelines for the use of animals [29,30].

### 2.8. Morphological Alteration

After the repellent bioassay, the morphological alterations to housefly antennae caused by the combination formulations were observed under a stereomicroscope (Nikon^®^ Model C-PS, Hollywood International Ltd., Bangkok, Thailand) and photographed with a digital camera (Nikon^®^ DS-Fi2, Hollywood International Ltd., Bangkok, Thailand).

### 2.9. Statistical Analysis

Statistical analyses were performed using IBM’s SPSS Statistical Software Package version 28 (Armonk, NY, USA). All bioassays were of completely randomized design (CRD). The choices, the landing on the treated or untreated filter paper, that the housefly had were analyzed using generalized linear models (GLM) with a binomial distribution. The tested EO was the explanatory variable, and the null hypothesis was that a housefly would choose either side of the test with equal probability [31]. Non-responding insects were not included in the analysis. The time that a substance took to produce 50% mortality (LT_50_) against non-target species with 95% confidence limits was determined by probit analysis on mortality (number of two non-target pollinators that had died at 72 h after exposure and number of two non-target aquatic predators that had died at 10 days post-treatment). Mortality data (±standard error) for the four non-target species and repellent rate were analyzed by one-way ANOVA, and Tukey’s test (*p* < 0.05) was used to investigate the differences across multiple treatment groups [32]. Repellent efficacy was analyzed by simple linear regression analysis. The main assumption of the simple regression analysis was checked. The assumption of linearity was true, with an R^2^ approaching one for all treatments (0.5323–0.9988).

## 3. Results

### 3.1. GC–MS Analysis of Two Plant Essential Oils

The qualitative and quantitative phytochemicals of lemongrass and star anise EOs were analyzed (Appendix A). The color of the two EOs was pale yellow. The lemongrass EO density was 0.97 ± 0.08 g per mL, and the star anise EO density was 0.98 ± 0.09 g per mL. The percentage of the extraction yield of star anise EO (4.01 ± 0.08% *v*/*w*) was much higher than that of lemongrass EO (1.21 ± 0.23% *v*/*w*). GC–MS analysis found that lemongrass EO was comprised of ten chemical constituents, accounting for 98.59 ± 0.87% of the total composition. Geranial (46.83 ± 1.02%) was the main constituent, followed by minor constituents such as neral, 1,8-cineole, geraniol, geranyl acetate, α-pinene, caryophyllene oxide, linalool, ϒ-terpinene, and α-terpinene. Respectively, the constituents’ peak areas were 23.85 ± 1.13%, 11.10 ± 0.95%, 6.10 ± 0.87%, 4.21 ± 0.93%, 3.25 ± 0.10%, 2.02 ± 0.52%, 0.88 ± 0.03%, 0.21 ± 0.02%, and 0.14 ± 0.02%. Star anise EO was comprised of seven constituents, accounting for 98.38 ± 0.98% of the total composition. The main constituent was *trans*-anethole (93.88 ± 1.05%), followed by minor constituents: limonene, *p*-anisaldehyde, 1,8-cineole, eugenol, α-terpineol, and α-thujene. Respectively, the peak areas were 1.84 ± 0.05%, 1.49 ± 0.05%, 0.81 ± 0.02%, 0.68 ± 0.03%, 0.43 ± 0.02%, and 0.25 ± 0.05%.

### 3.2. Repellent Activity

Figure 2 shows that all tested formulations provided a high repellent rate at 1–6 h after exposure, and the rate decreased with time after exposure. All single-component formulations were less effective (with a repellent rate between 33.8 to 94.4%) than all combinations (with a repellent rate between 55.1 to 100%). The highest repellent rate at 6 h after exposure among the eight single-component formulations was 70.0%, achieved by 1.0% geranial. The highest repellent rate among the seven combinations was 100% at 1–6 h of exposure, achieved by 1.0% geranial + 1.0% *trans*-anethole. In contrast, 12% *w*/*w* DEET gave a repellent rate of 66.7%, 52.4%, and 46.4% at 1, 3, and 6 h of exposure.

The effective repellent indexes (RI) of the tested formulations and 12% *w*/*w* DEET are shown in Figure 3. All combinations showed an RI in the range of 1.2 to 2.2. All of them were over two times more effective than all single-component formulations (RI ranging from 0.7 to 1.8). Among the eight single-component formulations, the strongest activity with the highest RI (1.5 to 1.8) was 1% geranial. It was 1.5 to 1.8 times more repellent than 12% *w*/*w* DEET. The lowest repellent activity was provided by 2.5% lemongrass EO and 2.5% star anise EO, with an SI of 0.7 to 0.8, about the same as that which 12% *w*/*w* DEET provided.

All combinations were 1.2 to 2.2 times (RI of 1.2–2.2) more highly repellent than 12% *w*/*w* DEET. Among the seven combinations, four formulations—2.5% lemongrass EO + 2.5% star anise EO, 0.5% geranial + 0.5% *trans*-anethole, 1.0% geranial + 1.0% *trans*-anethole, and 1.0% lemongrass EO + 1.0% *trans*-anethole—exhibited the strongest repellency, with 92.3 to 100% repellent rates at 6 h exposure. These four formulations were 2.0 to 2.2 times (RI of 2.0–2.2) more repellent than 12% *w*/*w* DEET. In particular, the combination of 1.0% geranial + 1.0% *trans*-anethole exhibited the greatest repellency, with a 100% repellent rate at 6 h of exposure time, which was 2.2 times (RI of 2.2) higher repellency than 12% *w*/*w* DEET.

The synergistic repellent index (SRI) and increased repellent value (IR) of the seven combinations (when compared with the single-component formulations) are shown in Figure 4. All combinations showed a highly synergistic effect with an SRI of 1.5 to 2.0. The greatest synergistic effect was achieved by the combination of 1.0% lemongrass EO + 1.0% *trans*-anethole, with an SRI of 2.0 and an IR of 51.2.

### 3.3. Toxicity to Non-Target Species

Toxicity against the two pollinators, dwarf honeybees and stingless bees, was evaluated. The 50% Lethal Times (h) against dwarf honeybees and stingless bees of all tested formulations are shown in Figure 5. All the tested single-component formulations and combinations were non-toxic to both pollinators, with an SI value ranging from 304.4 to 510.0. All the single-component formulations provided an SI value of 365.6 to 510.0 for stingless bees and 328.9 to 432.2 for dwarf honeybees. All the combinations provided an SI value of 315.0 to 361.3 for stingless bees and 304.4 to 372.2 for dwarf honeybees. In contrast, DEET exhibited high toxicity to both pollinators, with a low LT_50_ value of 0.08 to 0.09 h.

The 50% Lethal Times (days) against guppy and molly of all the tested formulations are shown in Figure 6. All the tested single-component formulations and combinations were safe to both aquatic predators, with an SI value ranging from 146.0 to 362.5. All the single-component formulations provided an SI value of 225 to 362.5 for molly and 194 to 298 for guppy. All the combinations provided a still-high SI value ranging from 205 to 325 for molly and 146 to 296 for guppy. In contrast, DEET was highly unsafe for both aquatic predators, with a low LT_50_ value of 0.04 to 0.05 days.

### 3.4. Morphological Alterations after Repellent Bioassay

After 6 h of exposure to each tested formulation, morphological changes of the antennae were observed. The shape of the antennae was abnormal, with sunken and twisted flagellum and aristae, as can be seen in Figure 7. This effect was especially pronounced with two combination formulations: 1.0% lemongrass EO + 1.0% *trans*-anethole and 1.0% geranial + 1.0% *trans*-anethole.

## 4. Discussion

Plant-based repellents are a safe option for preventing houseflies and other vector-borne diseases in epidemic areas [6,16,33]. Still, more studies are needed to improve green repellents in terms of efficacy, safety, and environmental friendliness [33,34]. Among plant products, plant EOs and their major constituents showed high potential as repellent agents against insect vectors [16,34]. The present study showed that lemongrass and star anise EOs and their active compounds were viable sources of phytochemicals that were highly repellent against houseflies. The EO yield and chemical composition of a plant EO strongly influenced its repellent property [16,19,34].

This study analyzed the EO yield and chemical composition of two plant EOs. The percentage of lemongrass EO yield in this study was 1.21 ± 0.23% *v*/*w*. This finding is in excellent agreement with some previous reports; lemongrass percentage EO yield was 1.1% *v*/*w* [16], 1.14% *v*/*w* [19], 0.5–1.13% *v*/*w* [35], and 1.01–1.46% *v*/*w* [36]. However, some other previous studies reported slightly different results: 0.21–0.29% *v*/*w* [37], 0.15–0.46% *w*/*w* [38], and 1.7% *w*/*w* [39]. As for star anise EO, its yield was 4.01 ± 0.08% *v*/*w*. This percentage yield was similar to those reported in many studies, such as 4.0% *v*/*w* [16], 4.0–4.5% *v*/*w* [40], and 4.13% *v*/*w* [41], but it differed considerably from the EO yield result reported by Soonwera et al. [42] at 9.6% *v*/*w.* The EO chemical analysis from the GC–MS of lemongrass and star anise revealed that their chemical compositions matched those previously reported [16,19,42,43]. Geranial was revealed to be the main constituent of lemongrass EO, accounting for 46.83 ± 1.02% of the total chemical composition. This percentage value was similar to those reported in several studies: the main compounds of lemongrass EO were geranial at 40.72% *v*/*w* [44], 42.42% *v*/*w* [45], 44.3% *v*/*w* [39], 45.40–45.41% *v*/*w* [16,19], and, slightly different from some other studies, geranial at 35.91% [46] and 51.14–53.21% *v*/*w* [47]. *Trans*-anethole (93.88 ± 1.05%) was the most abundant constituent of star anise EO. This finding is in good agreement with previous results [16,42] indicating that the *trans*-anethole percentage in the chemical composition was 93.58–94.0%. The findings from some other studies were slightly different: 80.0% [48], 81.04% [40], and 83.46% [41]. The difference in the EO yield and the chemical composition of lemongrass and star anise EOs were affected by many factors, such as plant genotype, plant age, maturity stage, harvesting stage, primary developmental period, quality of raw plant materials, and good management practice of cultivation [49,50,51]. The cropping time, stage of the plant, geographic distribution, environmental factors, and extraction and distillation methods all affected the quality and quantity of plant EOs [48,52]. Furthermore, lemongrass and star anise EOs and their major constituents have been demonstrated to possess some pharmacological activities (antimicrobial, antioxidant, anti-inflammatory, antibacterial, and antifungal). They have been used as food additives, preservatives, flavoring agents, perfumes, deodorants, and shampoos [48,53].

Regarding the efficacy improvement of plant EO-derived repellents, several studies suggested that combinations of EOs and their main constituents are synergistic [16,54]. They were more highly repellent than single-component formulations of the same strength [16,33]. The desired outcomes of a synergistic combination were to reduce the dose or concentration of the EOs in the formulation and to reduce or delay the risk of resistance development in insect vectors [34,42,55]. All combinations in this study exhibited increased repellent efficacy against houseflies, with a high repellent rate, an effective repellent index (RI), a high synergistic repellent index (SRI), and a high increased repellent value (IR). In particular, the combination of 1.0% lemongrass EO + 1.0% *trans*-anethole produced the strongest repellency with a high level of synergy, improving the repellent rate to more than 50%. Another outstanding formulation was the combination of 1.0% geranial + 1.0% *trans*-anethole, which gave the strongest repellent rate of 100% at all tested exposure times. Many studies reported that combinations of EOs and their main constituents were synergistically repellent. Geranium, lemongrass, and peppermint mixed with sunflower oil were highly repellent against horn fly (*Haematobia irrutans*: Diptera; Muscidae), with a repellent rate of more than 75% for 6–8 h [56]. A combination of lemongrass EO + peppermint EO (30:70) showed a synergistically strong repellency against houseflies, with an RC_50_ of 0.009 µ/m^3^ [15]. A combination of lemongrass EO + turmeric EO gave a synergistically high repellent activity against females of two mosquito species, *Anopleles minimus* and *Cx*. *quinquefasciatus*, with a protection time ranging from 120 to 125 min [57]. Combinations of lemongrass EO + olive oil and lemongrass EO + coconut oil showed high repellent activity against two species of mosquito females, *Ae*. *aegypti* and *Cx. quinquefasciatus*, with a protection time of 60–85 min and 115–170 min, respectively [18,58]. Moreover, some combinations of EOs or EO active compounds have been demonstrated to synergistically cause mortality against houseflies, such as a combination of 2.5% lemongrass EO + 2.5% eucalyptus EO [55]. A combination of 0.5% star anise EO + 0.5% geranial showed the strongest synergistic effect against houseflies, with 100% knockdown and mortality rates and an LT_50_ of 6.0 min [16]. Combinations of p-cymene + ϒ-terpinene, p-cymene + 1,8-cineole, and ϒ-terpinene + 1,8-cineole showed an acute synergistic adulticidal effect against houseflies [59].

On the mechanism of action, EOs and EO active compounds were not only neurotoxic, but their vapor damaged the antennae of mosquitoes and houseflies [11,60]. The vapor prevented an insect from landing on and biting humans and animals [60,61]. After the houseflies were treated with the combinations of 1.0% lemongrass EO + 1.0% *trans*-anethole and 1.0% geranial + 1.0% *trans*-anethole, their images were taken under a stereomicroscope and a digital camera. Morphological aberrations of the antennae were found, including abnormal antenna shape and sunken, twisted flagellum and aristae. Two combinations badly damaged the antennae of houseflies, hence interfering with and disrupting their smelling sense (Figure 7), so these combinations exhibited a high repellent rate for all tested exposure periods. This conclusion is similar to the finding in a study by Bakdacchino et al. [62]. They reported that lemongrass EO showed a repellent property against stable flies (*Stomoxys calcitrans*: Diptera; Muscidae). The mode of action was that the EO interfered with the antennal olfactory receptor cells. For mosquitoes (Diptera: Culicidae), their antennae, palpi, and tarsal sensilla are also rich in chemo-sensory receptors that are vulnerable to the vapor of an EO [61,62].

Plant EOs and their major constituents are mostly regarded as safe and eco-friendly. They are also generally considered safe for non-target organisms, such as pollinators, aquatic predators, and fishes [34,60,63,64,65]. In the present study, all the single-component formulations and combinations were safe to two pollinator species (dwarf honeybees and stingless bees) and two aquatic predators (guppy and molly), with a high SI and a high LT_50_ when compared to DEET. Two combinations, 1.0% lemongrass EO + 1.0% *trans*-anethole and 1.0% geranial + 1.0% *trans*-anethole, were outstandingly benign to both pollinators and aquatic predators, with an SI of 304.4 to 372.2 and 146.0 to 327.5, respectively. This finding is similar to the findings by Moungthipmalai et al. [19]. They reported that several combinations of cinnamon EO + geranial (2:1), lemongrass EO + D-limonene (2:1), citrus EO + geranial (2:1), D-limonene + geranial (1.5:1.5), geranial + *trans*-cinnamaldehyde (1.5:1.5), and D-limonene + *trans*-cinnamaldehyde (1.5:1.5) at 10,000 ppm were safe to non-target molly (*P*. *latipnna*) and guppy (*P*. *reticulate*), with a biosafety index (BI) of 1.06 to 2.57. Some EO compounds, such as thymol and 1,8-cineole, were moderately toxic to guppy, with an LC_50_ value of 10.99 to 12.51 mg/L and 1701.93 to 3997.07 mg/L, respectively [66]. Moreover, these biosafety results against pollinators were similar to the findings by Sahahi et al. [67]. They reported that lemongrass EO and *trans*-anethole did not show any toxicity to honeybees (*A*. *mellifera*), with an LD_50_ value of 53,304.0 and 35,942.0 µg/mL, respectively. Geranial, the main constituent of lemongrass EO also showed low toxicity to a predatory bug, *Podisus nigrispinus* [Heteroptera: Pentatomidae], with an LD_50_ value of 25.56 µg/insect [68]. On the other hand, DEET was highly toxic to the four non-target species, with a high mortality rate and a low LT_50_. Some studies reported that DEET showed high toxicity against fathead minnows (*Pimephales promeles*), resulting in a reduced number of androgen receptors in females [69]. It also affected non-target organisms in surface water systems and was highly toxic to some algae [70]. Although DEET has been classified by the United States Environmental Protection Agency (US.EPA.) as “Group D-not classifiable as a human carcinogenicity”, the safety of DEET for humans, the environment, and non-target organisms are of critical concern in many reports [71,72,73]. In contrast, EOs and their chemical compounds are eco-friendly, do not accumulate in the environment, and are safer for non-target organisms [19,60,61]. In this study, all the single-component formulations and combinations had high SIs, meaning that they were thoroughly safe for all tested non-target species. Moreover, lemongrass and star anise EOs and their main compounds have been used as food ingredients and folk medicine in many Asian countries since ancient times, and they did not alter the histopathology of rabbits, mice, and humans [47,48,74,75,76].

## 5. Conclusions

Two outstanding combinations, 1.0% lemongrass EO + 1.0% *trans*-anethole and 1.0% geranial + 1.0% *trans*-anethole, exhibited a highly synergistic repellency and a highly increased repellent value against houseflies at low concentrations and were safe for four non-target species. They should be developed further into a green repellent agent for managing housefly populations in houses, farms, and sensitive areas (nurseries, nursing homes, and hospitals). They are readily available, sustainable, and safe, which makes them an excellent option for inclusion into a repellent formulation that is safer and more effective than DEET, the existing popular housefly repellent. Further studies would include bioefficacy experiments on these two combinations, the development of stable formulations, and toxicity studies at the human cell level under laboratory and field conditions.

## Figures and Tables

**Figure 1 insects-15-00210-f001:**
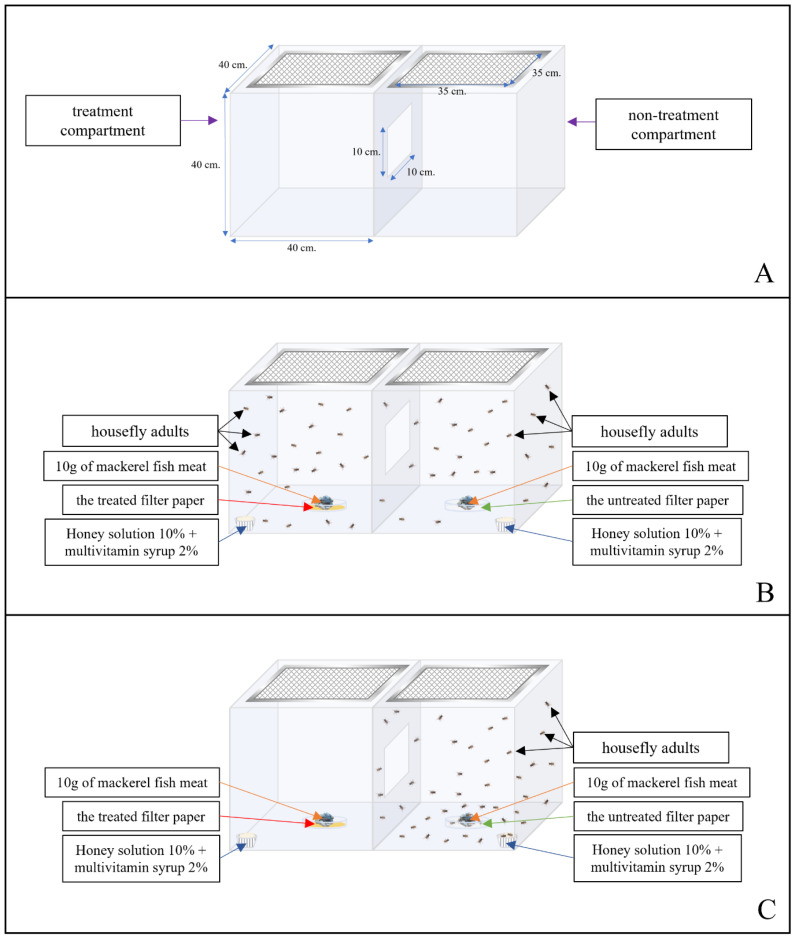
Repellent activity bioassay: The test cage was made of five plastic sheets (40 × 40 cm) and one screen sheet (35 × 35 cm) at the top of the cage, and the two compartments were connected by a rectangular hole (10 × 10 cm) in the middle of the cage (**A**). The first compartment on the left side contained a tested formulation, and the second compartment on the right contained the solvent (**B**). At the end, repellency could be observed and quantified (**C**).

**Figure 2 insects-15-00210-f002:**
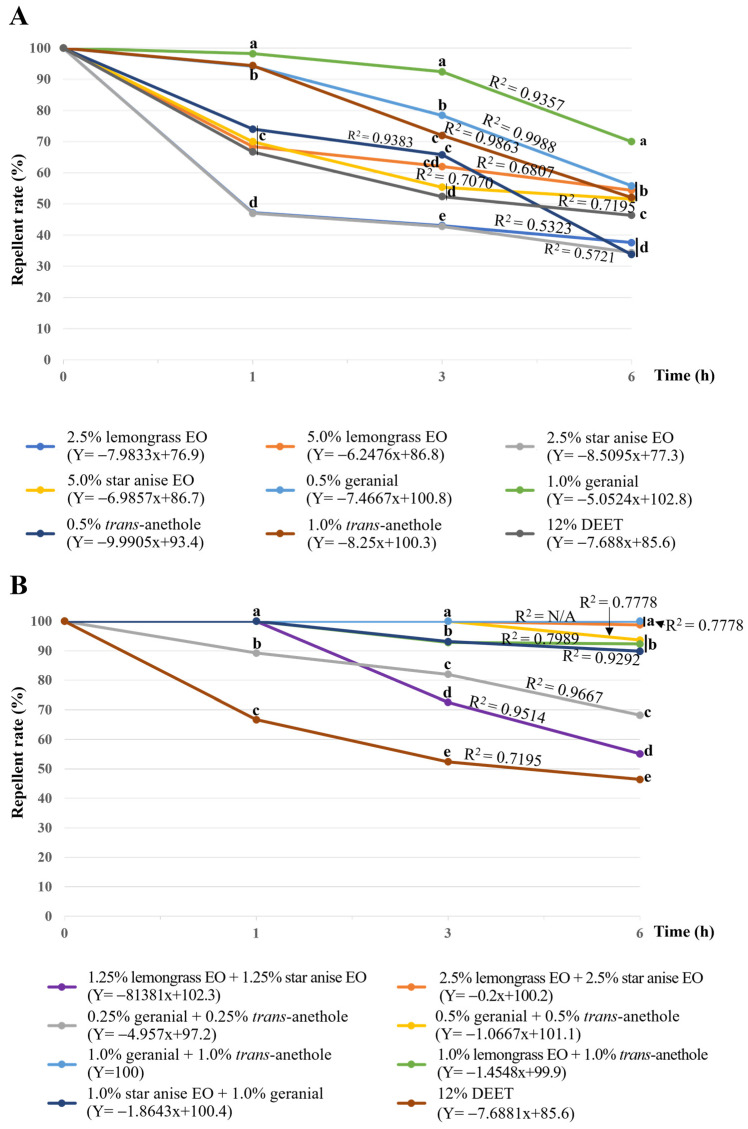
Repellent rate versus exposure time of formulations of lemongrass and star anise EOs and their main constituents: single-component formulations (**A**) and combinations (**B**). Note: Mean repellent rates within a row followed by the same letter do not differ significantly (Tukey’s post hoc test *p* < 0.05). N/A = not available.

**Figure 3 insects-15-00210-f003:**
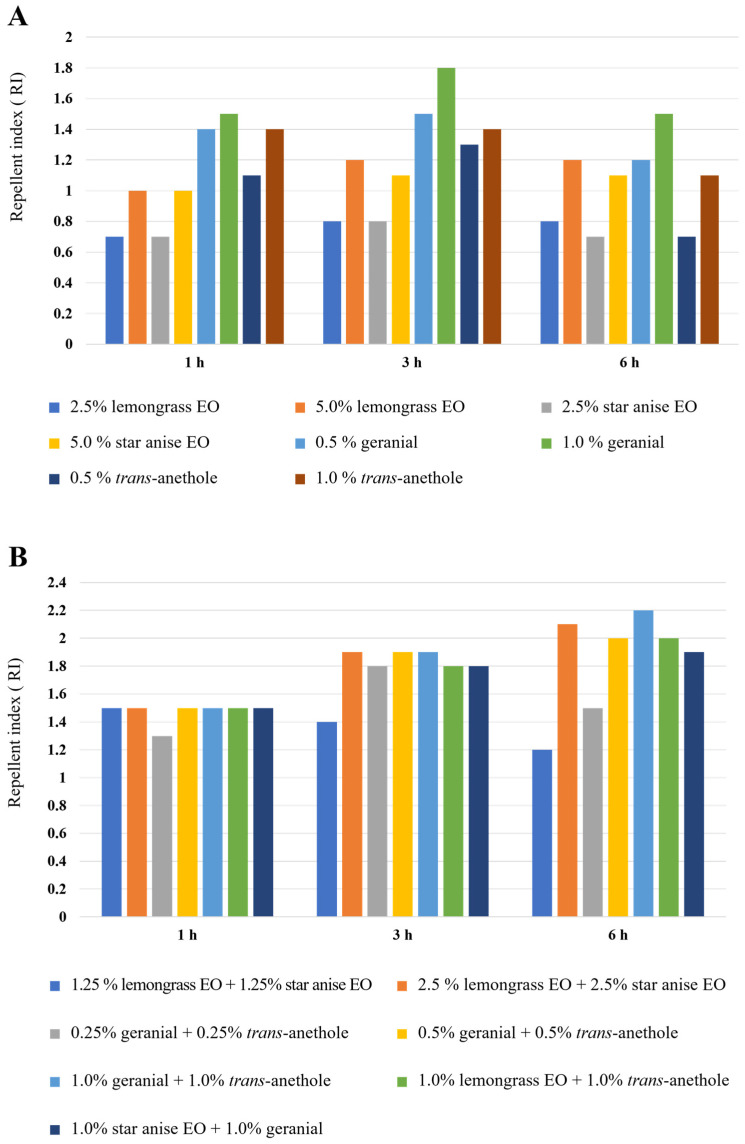
Repellent indexes (RI) of single-component formulations (**A**) and combinations (**B**). Note: RI is the ratio of the repellent rate of a single-component formulation or combination to the repellent rate of 12% DEET. A formulation with an RI < 1 means that the tested formulation was less toxic than DEET; a formulation with an RI > 1 means that the tested formulation was more toxic than DEET.

**Figure 4 insects-15-00210-f004:**
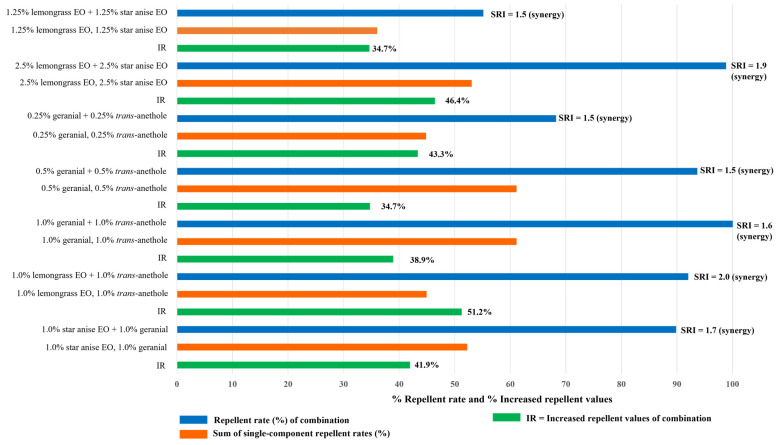
Synergistic repellent indexes (SRI) and increased repellent values (IR) of the combinations against housefly adults, when compared to the corresponding single-component formulations. Note: an SRI is calculated by dividing the repellent rate of the combination by the repellent rate of the single-component formulation. For combinations, an SRI < 1 means that the combination is synergistic; an SRI = 1 means that the combination was neither synergistic nor antagonistic; and an SRI > 1 means that the combination was antagonistic.

**Figure 5 insects-15-00210-f005:**
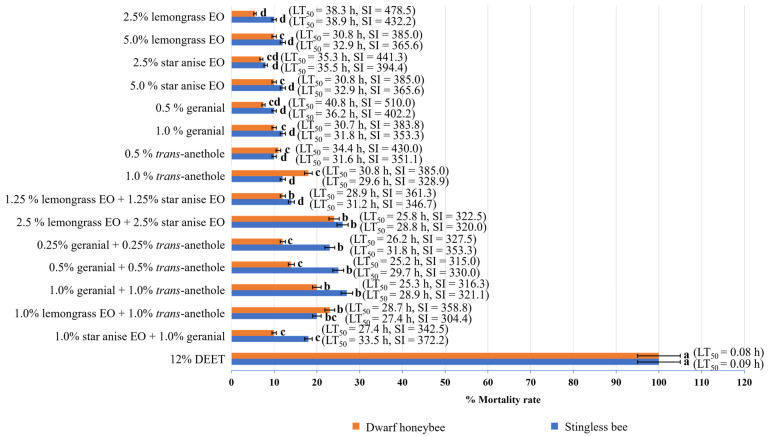
Mortality rates ±SE of single-component formulations and combinations against dwarf honeybees and stingless bees at 72 h after exposure. Note: Mean mortality rates ± SE within a row followed by the same letter do not differ significantly (Tukey’s post hoc test *p* < 0.05); LT_50_ = 50% Lethal Time; Safety index (SI): SI > 1 signifies that the formulation was safer than DEET for the non-target species, and SI < 1 signifies that the formulation was less safe than DEET for the non-target species.

**Figure 6 insects-15-00210-f006:**
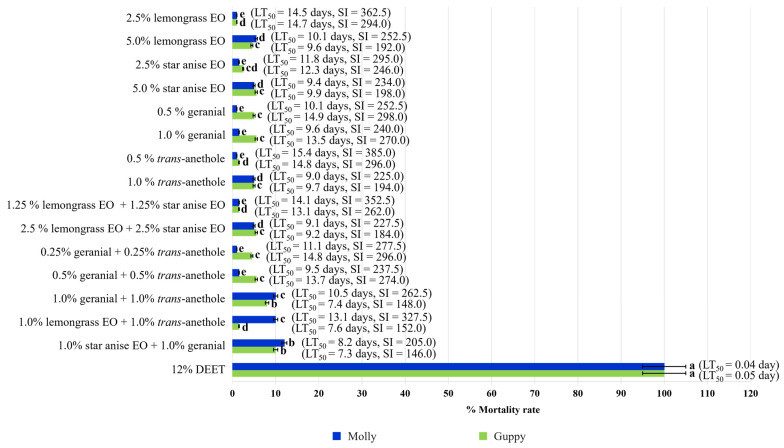
Mortality rates ±SE of single-component formulations and combinations against molly and guppy at 10 days post-treatment. Note: Mean mortality rates ±SE within a row followed by the same letter do not differ significantly (Tukey’s post hoc test *p* < 0.05); LT_50_ = 50% Lethal Time; Safety index (SI): SI > 1 signifies that the formulation was safer than DEET for the non-target species, and SI < 1 signifies that the formulation was less safe than DEET for the non-target species.

**Figure 7 insects-15-00210-f007:**
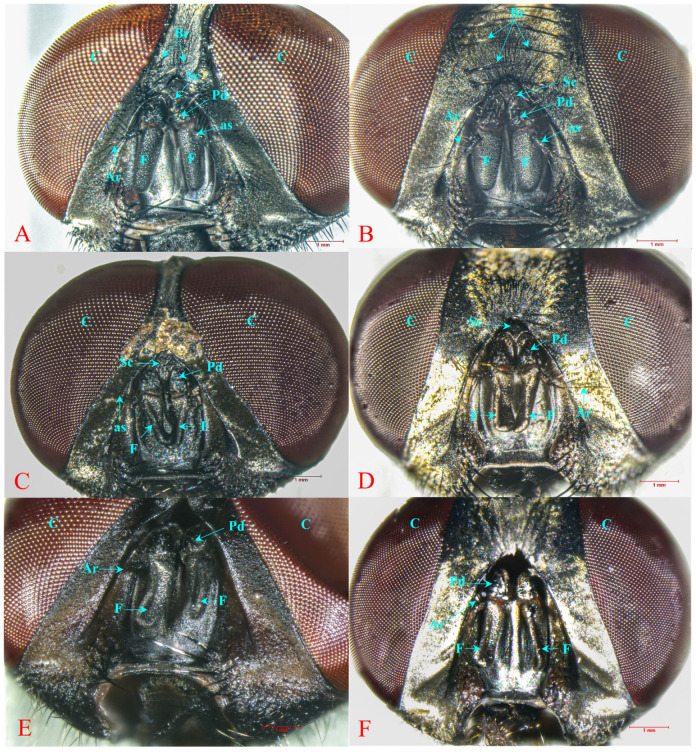
Light micrographs of housefly antenna: non-treated antenna, with normal shape of flagellum and aristae of female (**A**), male (**B**); abnormal shape and morphological damage of antenna, sunken and twisted flagellum and aristae after treatment with the combination of 1% lemongrass EO + 1% *trans*-anethole ((**C**)—female, (**D**)—male) and the combination of 1% geranial + 1% *trans*-anethole ((**E**)—female, (**F**)—male). Note: An antenna comprises a basal scape (Sc) and pedicel (Pd) with longitudinal antennal seam (as) and elongated flagellum (F), arista (Ar). Compound eye (C), bristle (Br).

## Data Availability

All relevant data are included in the article.

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
