# Peer review of "Combinations of Lemongrass and Star Anise Essential Oils and Their Main Constituent: Synergistic Housefly Repellency and Safety against Non-Target Organisms"

_insects, 2024, doi:10.3390/insects15030210_

Round 1
Reviewer 1 Report
Comments and Suggestions for Authors
To find environmental-friendly repellents for different disease-vector insects is crucial for sustainable environment, therefore the aim of the present manuscript is important and useful for the research community working on this objects, but also for a wider audience. The concrete subject of the study was to investigate the repellent activity of two essential oils and their main components alone or in combination with each other. For practical use it is important to find the optimal composition, ratio and amounts of the repellents. Investigating the effect of tested combinations on non-target organisms increase the value of the manuscript, as it is important to consider every aspect of the effectiveness of the potential repellence. Even the manuscript is a valuable work and I think it worth to be published, due to some unclarity in methodology and inappropriate aspects of results, I recommend publishing only after a major revision. More detailed comments can be found below.

Reviewer 2 Report
Comments and Suggestions for Authors
Combinations of lemongrass and star anise essential oils and their major constituent: synergistic housefly repellency and safety against non-target organisms. The aim of this study is to present study evaluated housefly repellency and repellent synergy of combined and individual formulations of lemongrass and star anise essential oils (EOs) and their active constituents. The paper is well written, but I do have some questions and concerns.
In order of the text and line numbers:
1) More details should be provided how EOs were chosen. What did preliminary data suggest>?
2) More details should be provided as why concentrations of EO single component and combination formulations were determined. What did preliminary data suggest?
3) How did you determine to use a concentration of 12% DEET? What does this concentration correspond to in terms of repellency?
4) Line 207- Line 207- please do not have the formula go over 2 lines.
5) Line 240- What concentration does 10,000 ppm correspond to in terms of concentration of repellent?
6) Lines 290-292- please check the order or ranking of compounds with respect to repellent rates. I believe that some of these compounds are not reported in the correct ranking order (e.g., 5% lemongrass should not be ranked ahead of 1% trans-anethole.
7) It is very difficult to follow the listing of repellent rates the way the authors have reported this currently. I would suggest putting this information into a tabular format AND labeling e.g., Figure 2 as Fig. 2A and Fig. 2B (Lines 308-311) to separate single versus combination formulations.
8) Why was DEET data not included in Fig. 2?
9) I would suggest including the repellent index values into a tabular format AND labeling e.g., Figure 3 as Fig. 3A and Fig. 3B (Lines 308-311) to separate single versus combination formulations.
10) Why was DEET data not included in Fig. 3?
11) I would suggest including data pertaining to synergistic repellent index and increased repellent values into tabular form.
12) The data shown in Fig. 4 is difficult to follow. Why is the data pertaining to SRI for compounds not listed in pairs e.g., SRI data for 1% lemongrass EO + 1% trans anethole and IR data for 1% lemongrass EO + 1% trans anethole.
13) Why was DEET data not included in Fig. 4?
14) The data presented in Figs. 5 AND 6 need to be re-written in bigger font. There is too much data, and it is impossible to read (e.g., standard error bars, letters to denote statistical levels, LT50, etc.).
15) The letters and arrows denoted in Fig. 7 are impossible to see and the photos are of poor quality, making it impossible to see any abnormal shapes in the antennae or morphological damage.
16) Putting data into tables (as per above) will help with making references in the Discussion section (Lines 399-516) and the Conclusions section (Lines 518-527).
Reviewer 3 Report
Comments and Suggestions for Authors
Overall, this is an interesting and valuable paper which should be published. There are, however, some clarifications and additions that need to be made. I will list them below.
For taxonomic clarity, and if supported by the journal, I suggest giving the family of each of the species mentioned in the study, including the non-target species.
Introduction mentiones repellency as well as insecticidal effects. Briefly give modes of delivery or application they are referencing in both cases - spray? bait? filter paper?
line 194. How were landings counted over time? How as the time spent in each cage recorded?
l. 213 vs 216. Mentions 100 vs 300 workers of each pollinator species - which is the correct number?
l. 231-233 Safety Index assumes DEET is safe for bees - is it? Clarify
l. 240 Starting concentration of 10,000 ppm in water is said to be equal to the concentration houseflies were exposed to in cages. How calculated? what other concentrations were tested?
l. 265 LT50 lethal time refers to what test substance? which concentrations? Describe the experiment and its design that produced the data for the probit anlysis, as this is very unclear.
Fig. 4. Comparing the response to combination formulations to the mean of the single components can be misleading, as the mean can obscure important differences between the single component formulations. Show the RI for each component separately, not their mean.
Fig 7. difficult to see rather obscure difference-- is there a better view of the flagellum and arista?
Reviewer 4 Report
Comments and Suggestions for Authors
The manuscript by Soonwera et al. describes a thoroughly undertaken study. Although I generally have no major concerns, the following points need to be addressed:
- line 70: I suggest to be more modest about whether EOs are the best option for repellent development.
- lines 104-105: remove sentence, because this paragraph should set out the aims and not a conclusion/judgement on EOs in general
- line 129: Was 100% EtOH used for dilutions before GC-MS?
- line 184: Were the 5 replicates done at the same time or at different timepoints? If the latter, was this considered for statistics?
- line 188: based on the figure, the interconnecting passage is a whole with only two dimensions, i.e. 10 x 10 cm
- line 215: provide reference for taxonomic identification
- lines 270-271: How was oil density measured?
- line 275: show TIC traces either in main text or in supplementary info
- lines 285-299: simplify paragraph by highlighting only major trends and not repeating info already shown in figures
- fig. 2-6: add significances
- lines 328-338: simplify paragraph by highlighting only major trends and not repeating info already shown in figures
- Fig. 7: symptoms not well-visible on photos, so I suggest you include a schematic to draw attention to important features
- lines 418-419: it is difficult to tell if geranial was the main active constituent in the EO; although it was active as a pure compound, it is in combination with other constituents in the EO with potential synergism amongst them
- line 511: Apart from lethal effects, were any data collected on other effects of EOs on bees, such as morphological defects?
Comments on the Quality of English LanguageThe English needs some polishing.
Round 2
Reviewer 1 Report
Comments and Suggestions for Authors
The resubmitted manuscript was significantly improved after my suggestions and those of the other reviewers were applied. The deficiencies were completed and the description of the work became more understandable and clear.
After reading the current version, I would only make a few small comments, akthought I think that the the publication process can go onward after a minor revision.
L. 216-218 - In my opinion, the relational signs are indicated here in reverse, SRI>1 would mean the significant effect (as can be read from Fig 4) and SRI < 1 the antagonist. I would also like to thank you for providing the descriptions of your equation with the appropriate abbreviations and notations, making the text much easier to interpret.
L. 319 - I found a small typo, '(' is not needed here
Reviewer 3 Report
Comments and Suggestions for Authors
The authors' responses to my questions are good as long as they incorporate the clarifications into the new version of the manuscript. There are a few remaining items that need to be clarified:
On line 266, it should be stated that an SI > 1 is safer than DEET, which is different than saying it is safe or benign. My original comment was that DEET was presumed to be safe, which authors say is not the case. This makes it even more important to clarify and clearly say that SI is a relative measure compared to DEET, and that an SI > 1 simply means the substance is relatively safer compared to DEET.
The count of fly landings is still unclear in lines 200-202. You now say "The number of landings for at most 5 minutes after the end of each period 200 were counted..." Why "at most 5 minutes"? It sounds very arbitrary, and counts for varying amounts of time up to 5 minutes will produce different results. Clarify or omit these data, as they sound quite unreliable.
My original question for line 240 was how an aqueous concentration of 10,000 pmm can be equated to the concentration tested in cages. Is it the concentration of the solution applied to filter paper in cages that is 10,000 ppm? If so a simple clarification would be good.
Assuming these changes and clarifications are made, I think the manuscript should move on to final review by editors.
Reviewer 4 Report
Comments and Suggestions for Authors
The authors have now addressed all my queries. However, I was not able to view the supplementary TIC traces (.rar files), so changing file extension might be needed.
Comments on the Quality of English LanguageSome minor edits to be done.
